# Efficient Pervaporation for Ethanol Dehydration: Ultrasonic Spraying Preparation of Polyvinyl Alcohol (PVA)/Ti_3_C_2_T_x_ Nanosheet Mixed Matrix Membranes

**DOI:** 10.3390/membranes13040430

**Published:** 2023-04-13

**Authors:** Huijuan Tong, Qiao Liu, Nong Xu, Qing Wang, Long Fan, Qiang Dong, Aiqin Ding

**Affiliations:** School of Energy, Materials and Chemical Engineering, Hefei University, Hefei 230601, China

**Keywords:** ultrasonic spraying, mixed matrix membranes (MMMs), composite membrane, MXene (Ti_3_C_2_T_x_), polyvinyl alcohol (PVA), pervaporation

## Abstract

Polyvinyl alcohol (PVA) pervaporation (PV) membranes have been extensively studied in the field of ethanol dehydration. The incorporation of two-dimensional (2D) nanomaterials into the PVA matrix can greatly improve the hydrophilicity of the PVA polymer matrix, thereby enhancing its PV performance. In this work, self-made MXene (Ti_3_C_2_T_x_-based) nanosheets were dispersed in the PVA polymer matrix, and the composite membranes were fabricated by homemade ultrasonic spraying equipment with poly(tetrafluoroethylene) (PTFE) electrospun nanofibrous membrane as support. Due to the gentle coating of ultrasonic spraying and following continuous steps of drying and thermal crosslinking, a thin (~1.5 μm), homogenous and defect-free PVA-based separation layer was fabricated on the PTFE support. The prepared rolls of the PVA composite membranes were investigated systematically. The PV performance of the membrane was significantly improved by increasing the solubility and diffusion rate of the membranes to the water molecules through the hydrophilic channels constructed by the MXene nanosheets in the membrane matrix. The water flux and separation factor of the PVA/MXene mixed matrix membrane (MMM) were dramatically increased to 1.21 kg·m^−2^·h^−1^ and 1126.8, respectively. With high mechanical strength and structural stability, the prepared PGM-0 membrane suffered 300 h of the PV test without any performance degradation. Considering the promising results, it is likely that the membrane would improve the efficiency of the PV process and reduce energy consumption in the ethanol dehydration.

## 1. Introduction

As a new type of bioliquid fuel, ethanol has been heralded all over the world as an alternative energy reducing air pollution and greenhouse gas emissions [1,2]. With aggravation of environmental pollution and exhaustion of fossil energy such as coal and crude oil, fuel ethanol has become an important energy source. The worldwide main source for fuel ethanol extraction is bio-fermentation. It can produce 6–10 wt.% ethanol, which is condensed to 95 wt.% by distillation [3]. However, the traditional distillation process cannot easily break the vapor-liquid equilibrium. Furthermore the reactive distillation and extractive distillation requires high energy consumption. Therefore, a new separation process is needed to further increase the concentration of the bioethanol, preparing anhydrous ethanol to replace the fossil energy. Pervaporation (PV) is a new type of membrane separation technology featured with low energy consumption, environmental friendliness, lower carbon-footprint, easy scale-up and easy coupling with other separation technologies [4,5]. Most importantly, based on the dissolution-diffusion model, it is less tightly bound by the vapor-liquid equilibrium, which endows it with advantages in difficult liquid separations such as azeotrope mixture and near-boiling point components [6]. In recent years, industrial applications of the PV processes in the field of organic solvent dehydration have been widely realized [7,8]. Many hydrophilic polymer membranes, such as chitosan (CS) [9], polyvinyl alcohol (PVA) [10], sodium alginate (SA) [11], etc. have been used for the separation of ethanol-water binary systems. The extraordinary physical and chemical properties of the PVA enable it to maintain the separation performance in long-term operation, which is suitable for providing basic membrane material during organic solvent dehydration [12]. In the 1980’s, the German GFT company fabricated commercial PVA composite membrane, realizing the commercial membrane application of the PV process [13]. Their PVA composite membrane with a flux of 1.0 kg·m^−2^·h^−1^ and separation factor of 100~200 could treat 80~95 wt.% ethanol aqueous solution. However, the large number of hydroxyl functional groups in the PVA molecular chains made it prone to swell in solution, resulting in defects in the membrane structure and limiting its further industrial application [14]. In order to improve the water resistance of the PVA membrane, thermal or chemical crosslinking method is employed to condense the hydroxyl functional groups among the PVA molecular chains, thereby inhibiting the swelling of the PVA membrane [15]. However, this will increase the density of the PVA membrane, reducing the free volume of the PVA molecular chains and the hydrophilicity of the membrane. Thus, it reduces the flux and separation factor of all PVA-based membranes. In order to improve the degree of the PVA crosslinking without sacrificing its hydrophilicity, many modification methods, for example grafting and blending, have been proposed [16,17]. Among them, blending with hydrophilic two-dimensional (2D) nanomaterials, such as MXene (Ti_3_C_2_T_x_-based), graphene oxide (GO), grafted (g)-C_3_N_4_, metal organic frameworks (MOFs), and covalent organic frameworks (COFs), has attracted extensive attention [18,19,20,21,22,23].

MXene is a new type of 2D material, which consists of a series of 2D transition metal carbides, carbonitrides and nitrides [24]. Atomic-thick MXene nanosheets can be obtained by selectively etching away the A-element atoms in the MAX phase ceramic material. For example, Ti_3_C_2_T_x_-based MXene nanosheets can be obtained by etching the Al atoms in the Ti_3_C_2_Al ceramic material [25]. Similarly to the GO, the surface and edge of the MXene nanosheets are rich in oxygen-containing functional groups, -OH and -O-, which endows it with excellent hydrophilicity. Furthermore, taking Ti_3_C_2_T_x_ nanosheet as an example, the single-layer of its nanosheet has a five-layer atomic structure which gives it excellent mechanical strength and thermal stability [26]. To date, the state-of-the-art PVA-based membranes have been modified by the MXene nanosheets for the PV process enhancement. Cai et al. incorporated the Ti_3_C_2_T_x_ nanosheets into the PVA matrix which promoted the crosslinking density of the PVA membrane and showed a significant increase in the PV performance [27]. Yang et al. prepared the Ti_3_C_2_T_x_ nanosheets with PVA to form the mixed matrix membranes (MMMs) which exhibited outstanding PV performances for various aqueous-ion or -alcohol mixtures dehydration [26]. Wang et al. and Li et al. added MXene nanosheets into natural polysaccharide sodium alginate and chitosan polymeric matrix to improve the PV performances of the MMMs [28,29]. So far, the researches on the MXene incorporated PVA-based MMMs for PV processes are still in the laboratory exploration stage. The large-scale preparations of the PVA/MXene MMMs for the PV processes with industrial applications are currently on hold.

Here in this work, we employed ultrasonic spraying method to fabricate a designed composite membrane. As shown in Figure 1, PVA-based casting solution with the Ti_3_C_2_T_x_ nanosheets as filler and glycerol as crosslinker was deposited on roll of poly(tetrafluoroethylene) (PTFE) electrospun nanofibrous support, followed with drying and thermal crosslinking sections. Then, the composite membrane with thin, homogenous and defect-free PVA-based separation layer on the PTFE supports was obtained. The prepared PVA composite membrane was characterized by Fourier transform infrared spectroscopy (FTIR), field emission surface scanning electron microscope with energy dispersive X-ray detector (FESEM-EDX), X-ray diffraction (XRD), thermogravimetry (TG), differential scanning calorimetry (DSC), X-ray photoelectron spectroscopy (XPS), and universal testing machine (UTM) to analyze its surface morphology, microstructure, chemical structure, crystal structure, thermal stability, and mechanical strength. The PV performances such as water flux and separation factor were also systematically tested by the homemade evaluation equipment. The membrane structure was optimized by the MXene nanosheets and its properties for the PV were improved significantly. The prepared PVA-based mixed matrix composite membranes showed excellent prospects for various industrial applications.

## 2. Experimental Section

### 2.1. Materials

Polyvinyl alcohol (PVA, purity ≥ 98.5%, alcoholysis degree ≥ 99%, Mw = 73,900–82,700 g·mol^−1^) was received from Anhui Wanwei Group Co., Ltd. (Chaohu, China). Glycerol (GC, purity ≥ 99.5%), LiF (Purity ≥ 99.99%), HCl (36–38%) and ethanol (AR grade) were purchased from Shanghai Titan Scientific Co., Ltd. (Chaohu, China). PTFE support (Fluoropore, width = 220 mm, length = 15 m, mean pore size = 0.22 μm) was purchased from Merck & Co. Inc. (Chaohu, China). MAX (Ti_3_AlC_2_) powder (400 mesh) was purchased from Nanjing Xianfeng Nanomaterials Technology Co., Ltd. (Chaohu, China). Deionized water (DI water) was homemade in the laboratory.

### 2.2. Preparation of Ti_3_C_2_T_x_-Based MXene Nanosheets

MXene nanosheets were fabricated by the in situ etching method which is provided in our previous study [30]. Briefly, 2.4 g LiF, 35mL HCl (12 M) and 2 g of MAX powder were mixed and stirred for 48 h at a temperature of 40 °C. After being diluted by and sonicated in DI water for 3 h, the mixture was filtrated and freeze-dried. The MXene nanosheets were received finally.

### 2.3. Preparation of the Composite Membranes

A certain amount (in Table 1) of the PVA and GC were added to boiling DI water. The mixture was continuously stirred until the PVA was completely dissolved. After cooling to room temperature, the PVA/GC solution was obtained. A certain amount (in Table 1) of the MXene powder was added in DI water at room temperature, and ultrasonically dispersed for 20 min to obtain a uniform MXene dispersion. The MXene dispersion was mixed with the PVA/GC solution and stirred at room temperature for 24 h to obtain a PVA/GC/MXene mixed solution. Finally, the PVA/GC/MXene casting solution was obtained after the mixture was left to stand for 24 h for defoaming.

The PVA/GC/MXene casting solution heated to 40 °C was added into the ultrasonic spraying system in Figure 1. The flow rate of the casting solution was adjusted to 10 mL/min. The N_2_ pressure was set at 0.05 MPa, the ultrasonic power was hold at 15 W, and the winding rate of the PTFE support (average pore size of 0.22 μm, an average thickness of 200 μm, a width of 350 mm, and a length of 30 m per roll) was set at 0.5 m/min. The temperature of the heating roll was set to 110 °C, and the temperature of the heating furnace was set to 150 °C. With the above operating parameters, fresh PVA/GC/MXene composite membrane was obtained using the equipment in Figure 1. According to different addition amounts of the PVA, GC and MXene, the prepared various PVA/GC/MXene composite membranes are listed in Table 1.

### 2.4. Characterization of the Membrane

Field emission surface scanning electron microscope with energy dispersive X-ray detector (FESEM-EDX, SU8010, Hitachi, Tokyo, Japan, operated at 10 kV) were used to analyze the microstructures of the membrane’s top-surfaces and cross-sections. Chemical structures of the membrane top-surfaces were investigated by FTIR (Nicolet iS 50+ Contiuum, Thermo Fisher, Waltham, MA, USA). X-ray photoelectron spectroscopy (XPS, Thermo escalab 250Xi, Waltham, MA, USA) was employed to analyze the surficial chemical structure of the membrane. Crystal structures of the membranes were examined by an XRD (TD-3500, Dandong Tongda, Dandong, China). TG and DSC analyses were conducted to analyze thermal property of the membrane (Mettler Toledo, Columbus, OH, USA, TGA/DSC 3+ series), where the carrier gas was high purity nitrogen (purity ≥ 99.999%). The temperature was increased from 30 °C to 700 °C with a heating rate of 10 °C·min^−1^ in the TG test, while the temperature ranged from 30 °C to 200 °C with a heating rate of 2 °C·min^−1^ during the DSC test. Water contact angle (WCA) was measured to characterize the hydrophilicity of all the membranes (KRUSS, DSA100).

### 2.5. Swelling Degree of the Membrane

Refer to the method described in the literature [28,31], the details are given as follows. The membrane was cut to a square with a size of 5 × 5 cm. Three pieces of the squares (5 × 5 cm) were weighed in dry state (m_1_) and soaked in 70 °C DI water for 48 h. Then the wet samples were taken out and weighed again (m_2_) after wiping off water droplets. The swelling degree (*SD*) of the membrane was calculated by Equation (1) with an average value of the three squares.
(1)$$SD=\frac{m_{2}-m_{1}}{m_{1}}\times 100\%$$

### 2.6. Tensile Property Test of the Membrane

The test method was provided in the literature [32,33]. The PVA composite membrane was cut into a dumbbell-like shape with length 115 mm, width 25 mm, and axis width 6 mm. The testing machine was adjusted with the test fixture spacing limit of 50 mm when fixing the sample on the machine. By using the TRAPEZIUM LITE X software (349-05249), the experimental parameters (both sample specifications and the data to be measured) were determined as follows. The tensile speed was set to 25 mm/min, the maximum load was 100 N, and the maximum stroke was 60 mm. The sample thickness was also input into the testing system with the relative humidity 50% and ambient temperature 25 °C. When the sample was pulled off, the experiment was terminated. Each sample was measured five times, and the average value was recorded. The tensile strength *T_s_* was calculated according to Equation (2).
(2)$$T_{s}=\frac{F}{S}$$
where *T_s_* is tensile strength (MPa), *F* is the maximum load when the samples were torn off (N), *S* is the cross-sectional area of the samples (m^2^).

At the same time, the elongation at break (E_b_) of the composite membrane was also measured.

### 2.7. Pervaporization (PV) Performance Measurement

A homemade PV testing device was used to characterize the fluxes and separation factors of all membranes, as shown in Figure 2 and Appendix A. The feed solution was the ethanol/water solution with ethanol concentration 95 wt.% at temperature 40 °C. The key PV parameters, total flux (*J*), water flux (*J_w_*), ethanol flux (*J_e_*), and separation factor (*α*) of the membrane were calculated by the following Equations (3)–(6) [34]
(3)$$J=\frac{W}{At}$$
(4)$$J_{e}=J\cdot y$$
(5)$$J_{w}=J-J_{e}$$
(6)$$\alpha =\frac{y/\left(1-y\right)}{x/\left(1-x\right)}$$
where *J* is the total flux (g·m^−2^·h^−1^), *W* is the mass of the permeate (g), *A* is the effective permeate area of the membrane (m^2^), *t* is the permeation time (h), *α* is the separation factor, *y* is the mass fraction of the ethanol in the permeate (g/g), *x* is the mass fraction of the ethanol in the feed solution (g/g). Here, the ethanol content in the permeate was detected by gas chromatography (GC-9860, Nanjing Haozhi Pu Analytical Equipment Co., Nanjing, China).

A long-term PV test was also conducted with the homemade PV device at 50 °C. After one hour circulating the feeding solution (95 wt.%, ethanol/water), the permeate compounds were collected every two hours in the cold trap. The key PV parameters could be calculated by Equations (3) to (6). To keep the composition and weight of the feeding solution unchanged, the permeate solution with the same concentration was added back in the feeding tank. The total operation time of the test was 300 h, including 9 h every night when the feeding side of the device was circulating normally and the vacuum on the permeate side was turned off without collecting the permeate.

### 2.8. Hansen Solubility Parameters

Hansen solubility parameter (HSP) were used to estimate the affinity between two materials [35]. *δ_d_* for dispersion, *δ_p_* for polarity and *δ_h_* for hydrogen bonding were the three components of the HSP. These three components, similar to three spatial coordinate axes, construct the Hansen solubility space. Any material and solvent can be found at exact coordinates in the Hansen solubility space. Thus, the solubility distance *R_a_* between any two materials can be calculated based on their spatial coordinates using Equation (7) [36].
(7)$$R_{a}=\sqrt{4{\left(\delta_{d1}-\delta_{d2}\right)}^{2}+{\left(\delta_{p1}-\delta_{p2}\right)}^{2}+{\left(\delta_{h1}-\delta_{h2}\right)}^{2}}$$

The radius *R*_0_ of the Hansen sphere of the material can also be calculated through its spatial coordinates [35]. The ratio of the solubility distance *R_a_* to *R*_0_ is defined as the relative energy difference (RED), as shown in Equation (8). The smaller the value of RED, the stronger the affinity between the two materials.
(8)$$RED=R_{a}/R_{0}$$

## 3. Results and Discussion

### 3.1. Microstructure and Chemical Structure

A single-layer Ti_3_C_2_T_x_ nanosheet was prepared by the in situ etching method of LiF and HCl [30]. The Ti_3_C_2_T_x_ nanosheets were freeze-dried thoroughly, and the grey powder was obtained, as shown in Figure 3a. Then, the aqueous dispersion of the grey powder was dropped on an anodic aluminum oxide (AAO) substrate and the dispersed nanosheets were observed under the FESEM. As shown in Figure 3b, the nanosheet with a radial dimension of 2 μm had a single-layered structure since the pore structure of the AAO underneath was observed clearly.

Spray-coating was a common method [37] for preparing composite membranes which could lower the thickness of the separation layer as much as possible. As shown in Appendix A, ultrasonic spraying generated a small droplet size, homogenous size distribution and had a soft impact on the support surface. Compared with the gas dynamic spray in Appendix A, it avoided the formation of air bubbles and promoted fabrication of a thinner and more homogenous layer on the support.

In Figure 4a–c, smooth and flat surfaces were observed in the PVA, PG and PGM-0 composite membranes. It demonstrated the homogenous coating of the ultrasonic spraying method. Several small bulges appeared on the surface (Figure 4c) of the PGM-0, and more Ti element can be observed in the top-surface of PGM-0 membrane than that of pure PVA membrane (Appendix A), which may ascribe to the existence of the Ti_3_C_2_T_x_ nanosheets. Additionally, in Figure 4d–f, the thicknesses of the separation layers were all about 1.5 to 2.0 μm without any defects between the separation layers and the PTFE supports. It demonstrated very good compatibility between the PVA-based separation layers and the PTFE supports.

As shown in Figure 5a, all of the PVA, PG and PGM-0 composite membranes had strong absorption peaks at the wavenumber of 3270 cm^−1^ and 1080 cm^−1^ which were ascribed to the C-OH structure of the PVA molecular chains. The peaks at the wavenumbers of 2920 cm^−1^ and 1408 cm^−1^ observed in the three composite membranes were ascribed to the C-H stretch and CH_2_ bend, which were mainly attributed to the carbon skeletal chain of each PVA molecular. No obvious difference existed between the FTIR curves of PVA and PG composite membranes because of the same elements and similar molecular structures of the PVA and GC molecules. However, the characteristic peak of the C-F structure was observed in the patterns of the PGM-0 composite membrane and pure MXene nanosheets. It obviously demonstrated the successful incorporation of the MXene nanosheets in the PGM-0 composite membrane [38]. More interestingly, as shown in Figure 5b, the characteristic peak of C=O in PGM-0 membrane shifted towards higher wavenumbers compared to PVA and PG membranes, indicating a strong hydrogen bonding interaction between the incorporated MXene nanosheets and PVA molecular chains [39]. At the same time, a weak characteristic peak of C-O-Ti appears at 710 cm^−1^ [40], indicating that some Ti-OH on the surface of MXene nanosheets has undergone crosslinking reaction with C-OH groups in the PVA molecular chains, generating C-O-Ti covalent bonds. Therefore, MXene nanosheets achieve crosslinking of PVA molecular chains through both hydrogen bonding interactions and covalent bonds. In addition, the characteristic peaks at 1235 cm^−1^, 1130 cm^−1^, 1038 cm^−1^, and 918 cm^−1^ correspond to the C-O-C structure [41], which is the covalent structure generated by crosslinking of GC molecules with PVA molecular chains and crosslinking between PVA molecular chains, and thus is commonly present in PVA, PG, and PGM-0 membranes.

In Figure 5b, the peak angle at 19.5° in the three PVA-based composite membranes was attributed to the semi-crystalline structures of the PVA polymeric matrix [42]. However, the Ti_3_C_2_T_x_ peak (001) [43,44,45] at the XRD pattern of the MXene powder was not observed in the PGM-0′s figure, implying that most of the Ti_3_C_2_T_x_ nanosheets were distributed in the bulk phase of the membrane and the small amount of the Ti_3_C_2_T_x_ nanosheets at the membrane’s top-surface did not excite obvious diffraction peak in the XRD pattern.

The response of XPS signal of the membrane chemical structure was sensitive when the composition of the membrane changed. As shown in Figure 6a, C, O and Ti elements were detected in the PGM-0 membrane’s top-surface. The relative content of Ti was 2.12 at.% in Table 2, implying the existence of Ti_3_C_2_T_x_ nanosheets in the PGM-0 composite membrane. Multi-peaks of C1s, O1s and Ti2p were resolved to the probable chemical structures as shown in Figure 6b–d. The C=O and C-OH were mainly derived from the GC and PVA molecules. The Ti-OH and C-Ti structures which were all originated from the Ti_3_C_2_T_x_ nanosheet were observed in the multi-peaks of C1s, O1s and Ti2p. According to the relative content of the C-Ti structure in the C1s, 6.06 at.% in Table 2, it was confirmed that few MXene nanosheets were distributed on the top-surface of the PGM-0 composite membrane and most of them were incorporated in the membrane’s bulk phase.

Based on the analysis above, we can infer that MXene nanosheets in the membrane mainly achieve crosslinking with PVA molecular chains through hydrogen bonding interactions between their surface Ti-OH and C-OH groups in PVA molecular chains, as well as Ti-O-C covalent bonds. GC molecules mainly achieve crosslinking with PVA molecular chains through covalent bonds. Therefore, GC and MXene nanosheets will significantly change the performances and structures of PVA membrane.

### 3.2. Thermal and Mechanical Stability

As a composite material, both the thermal and mechanical stabilities of the membrane were important for its potential industrial application in the future.

In Figure 7a, according to the heat flow curves of the three membranes, glass transition temperature (*T_g_*) of the pure PVA membrane was only 79.1 °C, and the *T_g_* of the PG and PGM-0 membranes were increased to 92.8 °C and 101.4 °C, respectively, after the crosslinking with the GC molecules and the incorporation of the MXene nanosheets. When the ambient temperature increased to 230 °C and 400 °C, a great mass decrease was observed in the PVA and PG membranes in Figure 7b, which mainly originated from the degradation of the PVA polymeric matrix [46,47]. The initial degradation temperature and mass retention rate of the PG membrane were a little higher than those of the PVA membrane, implying that the crosslinker GC molecules enhanced the thermal stability of the PVA matrix. Furthermore, the weight loss curve of the PGM-0 membrane was slightly different from those of the other two membranes. The weight loss in the temperature range of 300 °C to 350 °C mainly came from the degradation of the MXene nanosheets. The initial degradation temperature of the PVA polymeric matrix and mass retention rate in the PGM-0 membrane curve were obviously higher than those of the PVA and GC membranes, demonstrating that the MXene nanosheets in the PGM-0 membrane greatly improved the thermal stability of the PVA matrix. Same as the present result, Pan et al. [48] also proved that the MXene improved the thermal stability of the PVA matrix. In Woo’s [49] work, the reason the thermal stability increasing the MXene/PVA nanocomposite was attributed to the excellent interfacial interaction between the MXene and PVA via crosslinking. Hence, the polymeric PVA matrix could be crosslinked by the GC molecules and MXene nanosheets (as shown in Figure 8). Both of them reinforced the PVA-based membrane matrix and made the membrane structure stable later on. Based on the above analysis, as shown in Figure 7c, it is reasonable that the swelling degree (*SD*) of the PGM series membrane was mainly lower than those of the PVA and PG membranes. As in the PGM series membranes, the membrane’s *SD*s were influenced by the relative contents of the PVA, GC and MXene. When the relative contents of the GC and MXene decreased (Figure 7c), the *SD* increased. Reversely, the relative contents of the GC and MXene increased (Figure 4d,e). The *SD* decreased. This was also in accord with their influence on the structure and stability of the PVA-based membranes. Moreover, when the addition amount of the MXene was 3 g (Figure 7e), the *SD* increased a little, which was attributed to the inhomogeneous distribution and agglomeration of the MXene nanosheets in the PVA polymeric matrix.

As in the WCA, values of the PGM series were lower than those of the PVA and PG membranes, indicating the enhancement of the membrane hydrophilicity. It was mainly attributed to the addition of the MXene nanosheets in the PVA matrix. Meanwhile, from Table 3, it can be seen that the RED value of MXene and water was much lower than that of PVA and water, indicating that MXene has a stronger affinity with water. This is also the main reason for the reduction in the water contact angle of the PGM membrane. Although both of the GC and MXene nanosheets had crosslinking effects on the PVA molecule chains, their crosslinking mechanisms were different. Similarly to the GO, the MXene nanosheets improved the stability of the PVA matrix mainly through the hydrogen bonding between hydroxyl groups on its surface and hydroxyl groups on the PVA molecular chains [48,49]. However, the connection of the GC molecules and PVA molecule chains were mainly the covalent bounds generated between the carboxyl in the GC molecules and the hydroxyl in the PVA molecular chains. It resulted in the loss of -OH groups in the PVA matrix and reduced the membrane hydrophilicity eventually. Hence, crosslinking of the GC molecules had two-way effects on both the WCA and the *SD* of the membrane. On the contrary, the added MXene nanosheets promoted the membrane hydrophilicity. Moreover, when the addition amount of the MXene reached 3 g, the influence of the MXene on the membrane hydrophilicity was limited.

Mechanical properties of the composite membranes were also measured. As shown in Figure 9, additions of the PG and MXene nanosheets were beneficial to the improvement of the tensile strength but detrimental to the enhancement of the elongation at break. Tensile strength of the PVA composite membranes had positive correlation with the crosslinking of the PVA polymeric matrix. However, increasing the strength of the PVA polymer matrix caused a decrease in the ductility of the composite membranes, resulting the decline in the elongation at break.

On the whole, both the GC and MXene could improve the crosslinking of the PVA membranes, thereby enhancing the membrane thermal stability and mechanical strength. However, adding GC consumed the hydroxyl group in the PVA matrix and reduced the hydrophilicity of the PVA membranes. Meanwhile, MXene supplied the oxygen-containing functional groups to the PVA membrane matrix and improved the hydrophilicity of the PVA membrane. Nevertheless, adding excessive MXene intrigued agglomerate in the PVA matrix, leading the anti-swelling property and hydrophilicity of the membrane worsen. Therefore, the PGM-0 with appropriate loadings of the GC (5.0 g) and MXene (1.0 g) has the best thermal stability, anti-swelling, hydrophilicity, and mechanical properties in this study.

### 3.3. Pervaporation Experiment

As shown in Figure 10, the pure PVA membrane had low PV performance; its water flux and separation factor were only 0.26 kg·m^2^·h^−1^ and 125.4, respectively. After crosslinking with the GC, the crosslinked PVA network was constructed. Free volume of the membrane matrix and the hydrophilicity decreased. It led to the fall in the water flux but increase in the separation factor. When the MXene nanosheets were incorporated in the PVA matrix, both its water flux and separation factor were increased to 1.21 kg·m^2^·h^−1^ and 1126.8 simultaneously.

According to the solution-diffusion model [50,51,52], water and ethanol molecules were all dissolved in the membrane top-surface first, then diffused to the permeation side of the PVA MMMs under the concentration gradient. Due to the better affinity of the water molecules with PVA MMMs than that of the ethanol molecules, water molecules were preferentially dissolved into the membrane top-surface and diffused to the permeation side, which resulted in the ethanol dehydration in the feed side. The incorporation of the MXene nanosheets in the PVA matrix improved hydrophilicity of the MMMs and enhanced the affinity for the water molecules (shown in Table 2) [27], resulting in the preferential dissolution of water molecules in the membrane top-surface and eventually promoted the selectivity (separation factors). Meanwhile, uniformly dispersed MXene nanosheets in the PVA membrane matrix formed phase interfaces with the PVA matrix. It constructed hydrophilic permeate channels for water molecules, and accelerated their diffusion speed [28]. All of these properties significantly improved the water flux of the membrane. More importantly, ultrasonic spraying preparation in Figure 10 promoted the fabrication of the thin and homogenous separation layer on the PTFE support. The method had great advantages in homogeneity and industrial application than other preparation methods such as tape-casting and spin coating.

The influences of the addition amount of the PVA, GC and MXene on the membrane’s PV performances were also investigated in Figure 11 and Appendix A.

As shown in Figure 11a, water flux and separation factor of the PGM series MMMs were all increased, then decreased as the PVA addition amount increasing. When the PVA addition was 10 g (PGM-1), the GC amount was excessive, leading to the over-crosslink of the PVA molecular chains. Hence the PV performance was limited. When the PVA content was higher than 30 g, the PV performance of the PGM MMMs were decreased instead, which was ascribed to the increasing membrane thickness and the decreasing membrane crosslinking degree. Low GC content led to low crosslinking degree of the PVA membrane matrix, while high GC content caused over-crosslink of the PVA membrane matrix, as shown in Figure 11b. Therefore, the best PVA content and GC addition amount were 30 g and 5 g in this study. As for the MXene, the nanosheets greatly improved the hydrophilicity of the PVA-based membrane matrix, inducing the performance enhancement during the PV process. However, the overloaded MXene nanosheets agglomerated in the membrane matrix, resulting in a decrease in the water flux and separation factor. Hence, in Figure 11c, water flux and separation factor both declined when the MXene loading was over 1.0 g. After careful analysis of Figure 11a–c, it was concluded that the most popular MMM was PGM-0 composite membrane. As a result, long-term PV test was conducted for the PGM-0. In Figure 11d, water flux and separation factor, that kept stable in 300 h of the PV test, were 1.2 kg·m^2^·h^−1^ and 1120, respectively, indicating that the fabricated PGM-0 composite membranes by the ultrasonic spraying method had a stable structure without breaks and/or defects generation during the long-term operation. Furthermore, compared to the PVA MMMs reported in the other literature in Figure 12 [47,53,54,55,56,57,58,59,60,61,62], the PGM-0 membranes had better water flux than others.

Although the separation factor of the PGM-0 was not the best in Figure 12, the reported high separation factor was mainly attributed to the additions of the nano-size molecular sieving materials in Appendix A. Both nanomaterials (the MXene and molecular sieve) have different advantages according to their own properties. Further research is necessary in their PV process. In this study, considering several factors such as PV performance, long-term operation stability and the advancement of the preparation method (ultrasonic spraying), the PGM-0 composite membrane has broad application prospects in the field of the ethanol dehydration.

## 4. Conclusions

In this work, PVA-mixed matrix composite membrane for the PV process of ethanol dehydration was prepared by the homemade ultrasonic spraying equipment for the first time. A thin, homogenous, and defect-free PVA-based separation layer was perfectly coated on the PTFE support by an ultrasonic nozzle, and rolls of the PVA composite membranes were fabricated by the following process of drying and thermal crosslinking. The prepared PVA composite membranes were investigated systematically. Both the crosslinking agent (glycerol, GC) and the MXene nanosheets could improve the mechanical strength and the crosslinking degree of the membrane. However, GC consumed the hydroxyl functional groups in the PVA matrix, thus reducing the hydrophilicity and pervaporation performance of the membrane. The MXene nanosheets supplemented the hydroxyl functional groups in the PVA matrix, enhancing the hydrophilicity of the membrane instead. The PV performance of the membrane was significantly improved by increasing the solubility and diffusion rate of the membrane to the water molecules through the hydrophilic channels constructed by the MXene nanosheets in the membrane matrix. The water flux and separation factor of the prepared PGM-0 mixed matrix composite membranes was dramatically increased to 1.21 kg·m^−2^·h^−1^ and 1126.8, which had advantages over other reported PVA-based MMMs in the references. Moreover, with the high values of tensile strength (70.6 Mpa), elongation at break (4.1%), and anti-swelling property (the *SD*, 43%), the PGM-0 membrane had a strong capability of maintaining permeation performance and membrane structure stability during a 300 h PV test. Generally, the prepared PGM-0 composite membrane had broad industrial application prospects in the field of the ethanol dehydration.

## Figures and Tables

**Figure 1 membranes-13-00430-f001:**
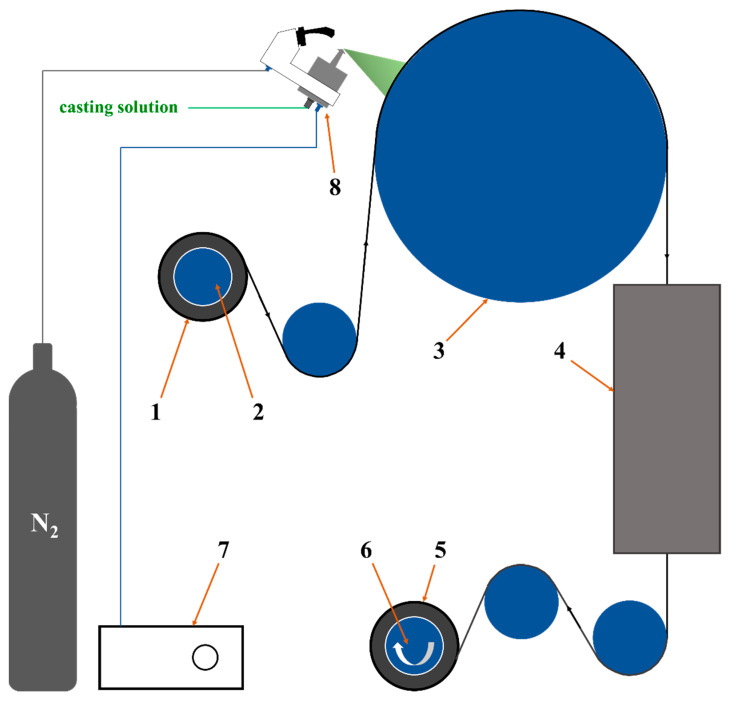
Schematic diagram of preparing the PVA composite membrane by homemade ultrasonic spraying equipment. 1. PTFE supports 2. Unwinding roller 3. Heating roller 4. Heating furnace 5. PVA composite membrane 6. Winding roller 7. Ultrasonic spraying controller 8. Nozzle.

**Figure 2 membranes-13-00430-f002:**
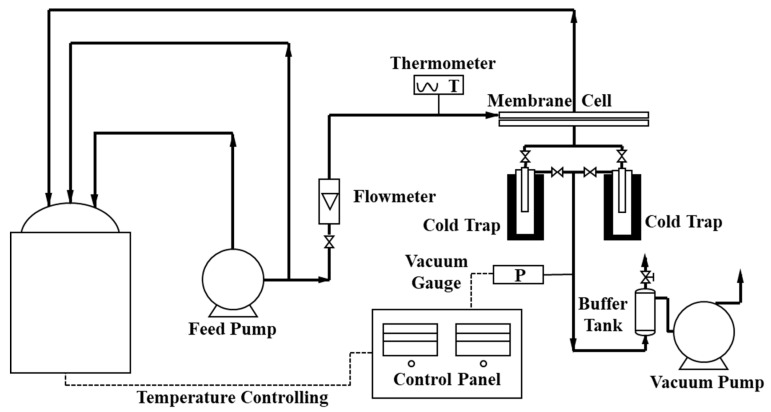
Flow diagram of the homemade PV device.

**Figure 3 membranes-13-00430-f003:**
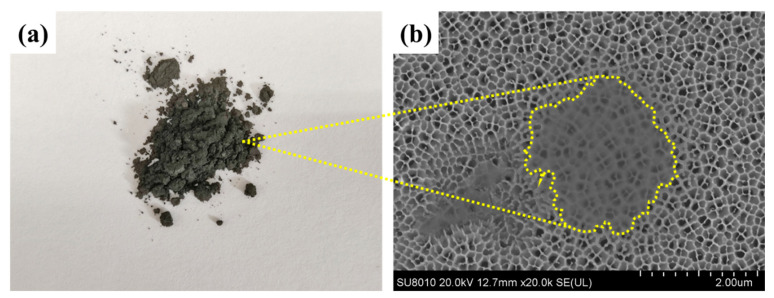
(**a**) MXene (Ti_3_C_2_T_x_) powder, (**b**) FESEM image of the MXene nanosheet on an anodized aluminum oxide (AAO).

**Figure 4 membranes-13-00430-f004:**
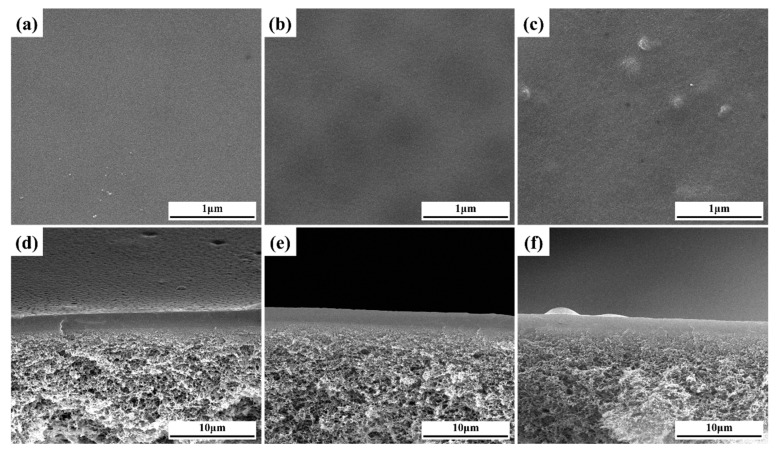
(**a**–**c**) top-surfaces and (**d**–**f**) cross-sections of the PVA, PG, and PGM-0 composite membranes.

**Figure 5 membranes-13-00430-f005:**
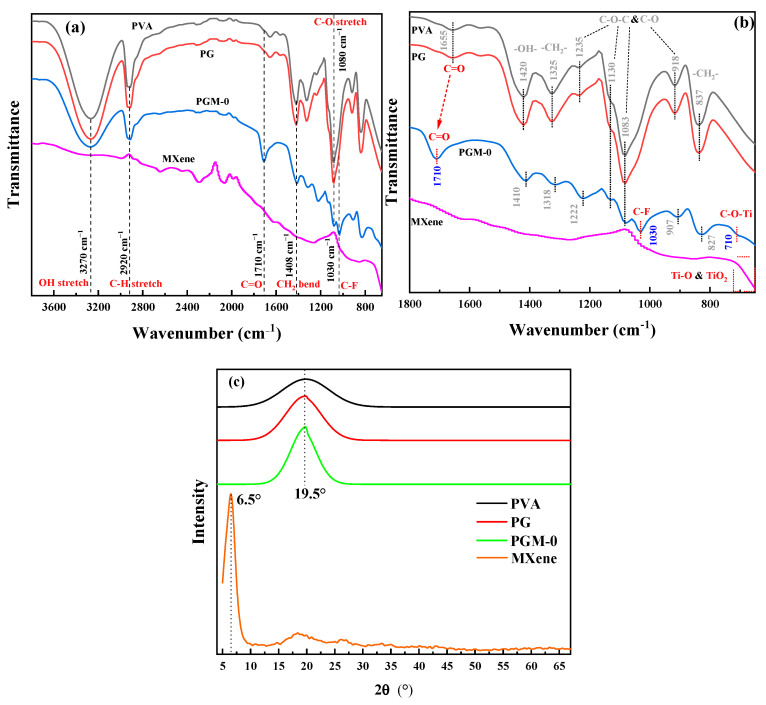
(**a**,**b**) FTIR spectra and (**c**) XRD curves of the PVA, PG, PGM-0 composite membranes and the MXene powder.

**Figure 6 membranes-13-00430-f006:**
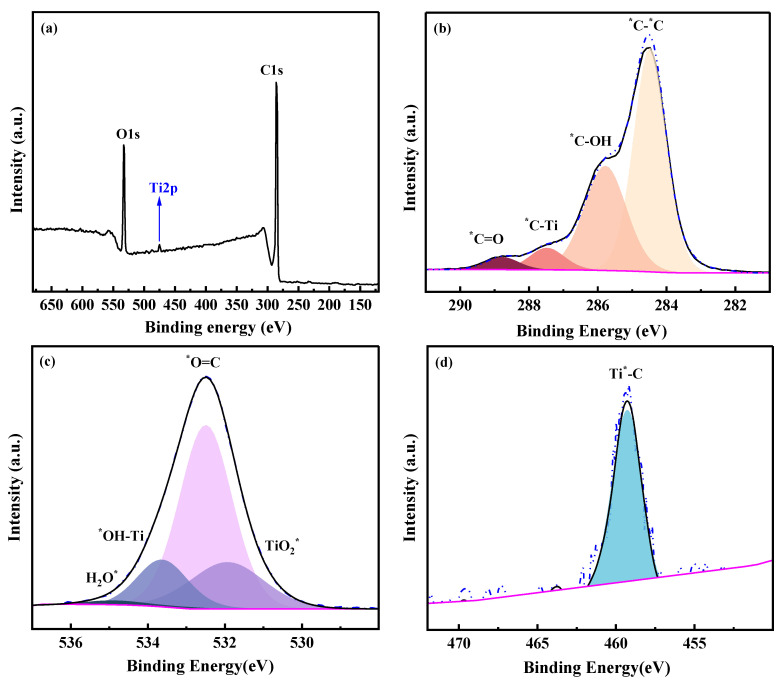
(**a**) XPS spectra and XPS peak resolutions of (**b**) C1s, (**c**) O1s, (**d**) Ti2p of the top-surface of the PGM-0 membrane. * represents the elements with partial peaks.

**Figure 7 membranes-13-00430-f007:**
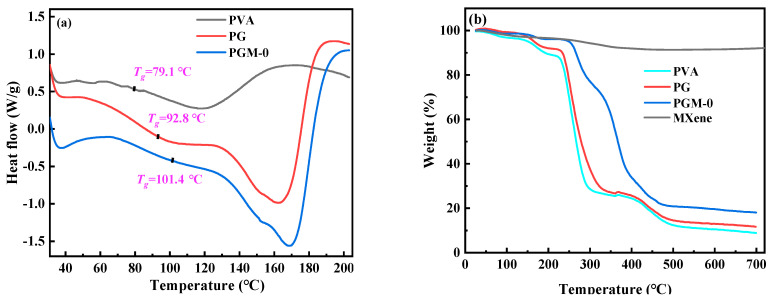
(**a**) DSC curves of the PVA, PG, PGM-0 membranes, (**b**) TGA curves of the PVA, PG, PGM-0 membranes and pure MXene, swelling ratios and water contact angles of the PVA MMMs with different addition amount of the (**c**) PVA, (**d**) GC and (**e**) MXene listed in Table 1.

**Figure 8 membranes-13-00430-f008:**
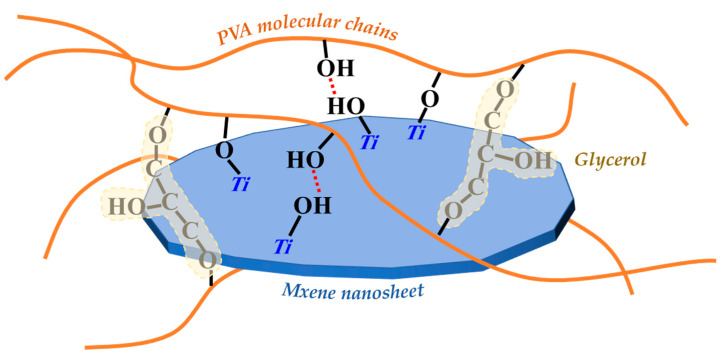
Schematic diagram of the crosslinking of PVA molecular chains, GC molecules and MXene nanosheets.

**Figure 9 membranes-13-00430-f009:**
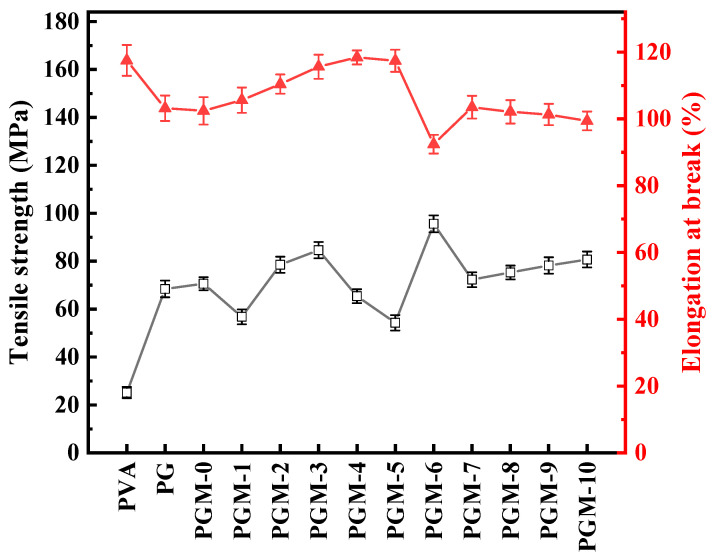
Mechanical properties of the as-prepared membranes.

**Figure 10 membranes-13-00430-f010:**
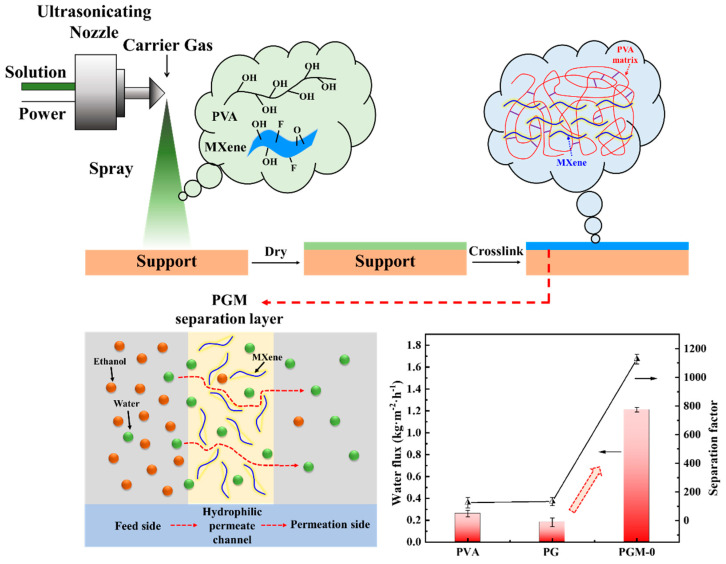
Fabrication of the hydrophilic permeate channels in the PVA membrane matrix and its mechanism for ethanol/water separation.

**Figure 11 membranes-13-00430-f011:**
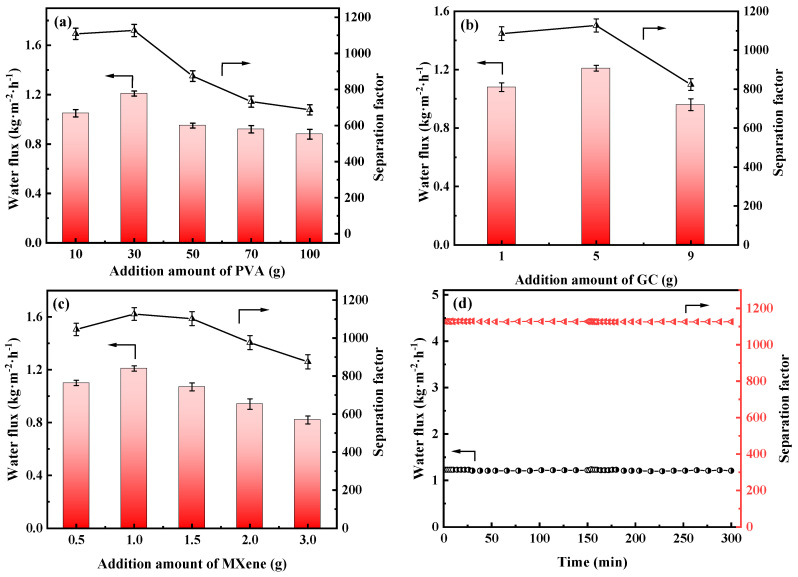
Water flux and separation factor of the PVA MMMs with different addition amount of the (**a**) PVA, (**b**) GC and (**c**) MXene, (**d**) long-term stability of the PGM-0 composite membrane.

**Figure 12 membranes-13-00430-f012:**
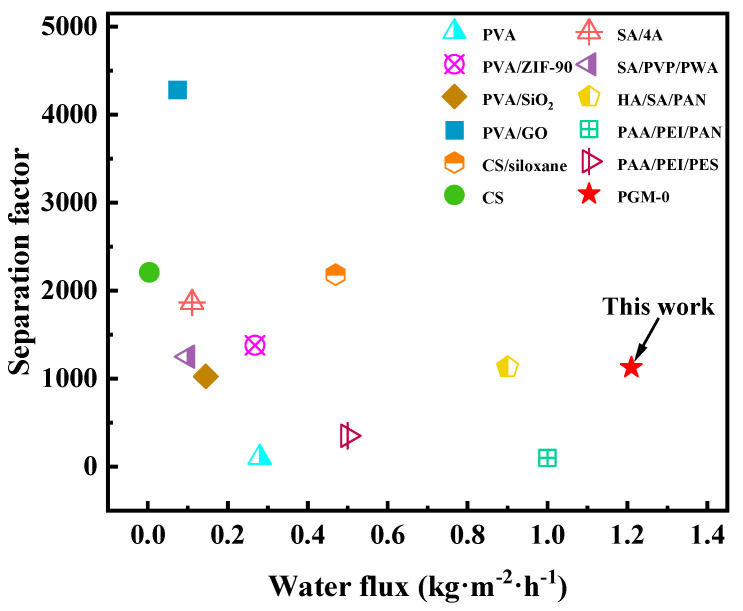
Comparison of the PV in this study with others.

**Table 1 membranes-13-00430-t001:** The composition of different PVA/GC/MXene composite membranes.

Membranes (Dense Separation Layer)	Addition Amounts (g)				Support (0.22 μm Pore Size)
	PVA	GC	MXene	Water	
PVA	30	0	0.0	970.0	PTFE
PVA/GC (PG)	30	5	0.0	965.0	PTFE
PVA/GC/MXene (PGM-0)	30	5	1.0	964.0	PTFE
PVA/GC/MXene (PGM-1)	10	5	1.0	984.0	PTFE
PVA/GC/MXene (PGM-2)	50	5	1.0	944.0	PTFE
PVA/GC/MXene (PGM-3)	70	5	1.0	924.0	PTFE
PVA/GC/MXene (PGM-4)	100	5	1.0	894.0	PTFE
PVA/GC/MXene (PGM-5)	30	1	1.0	968.0	PTFE
PVA/GC/MXene (PGM-6)	30	9	1.0	960.0	PTFE
PVA/GC/MXene (PGM-7)	30	5	0.5	964.5	PTFE
PVA/GC/MXene (PGM-8)	30	5	1.5	963.5	PTFE
PVA/GC/MXene (PGM-9)	30	5	2.0	963.0	PTFE
PVA/GC/MXene (PGM-10)	30	5	3.0	962.0	PTFE

**Table 2 membranes-13-00430-t002:** XPS binding energy and relative content of each element (C, O, S and Ti) and chemical structures originated from the peak resolution.

Elements	C1s				O1s				Ti2p
Structures	C-C	C-OH	C-Ti	C=O	O=C	HO-Ti	TiO_2_	H_2_O	Ti-C
Peaks (eV)	284.78	285.78	287.48	288.78	532.49	533.64	531.91	534.78	459.28
Relative Content (at.%)	58.00	34.99	6.06	1.22	61.42	14.82	21.26	2.50	100.00
	71.24				26.64				2.12

**Table 3 membranes-13-00430-t003:** Hansen’s solubility parameters for the PVA and MXene (Ti_3_C_2_Tx) and distance parameter (*R_a_*) and RED calculated according Equations (7) and (8).

	*δ_d_* (MPa^0.5^)	*δ_p_* (MPa^0.5^)	*δ_h_* (MPa^0.5^)	*R_0_* (MPa^0.5^)	*R_a_*	RED
PVA	17.0	9.0	18.0	4.0	25.46	6.37
Ti_3_C_2_T_x_	18.7	15.4	14.5	11.0	28.53	2.59
Water	15.5	16.0	42.3	-	-	-

## Data Availability

Not applicable.

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
