# Peer review of "Efficient Pervaporation for Ethanol Dehydration: Ultrasonic Spraying Preparation of Polyvinyl Alcohol (PVA)/Ti3C2Tx Nanosheet Mixed Matrix Membranes"

_membranes, 2023, doi:10.3390/membranes13040430_

Round 1

Reviewer 1 Report

This manuscript discusses the modification of PVA-based membranes by MXene and subsequently application of MMMs for the dehydration of ethanol. This manuscript is well organized and just needs a few minor improvements.

 I have some comments that should be addressed in order for the manuscript could be considered for publication.

1. Experimental section.

- Why did the authors not prepare membranes using a constant casting solution flow rate and PTFE support winding rate? The range of flow and the winding rate is listed in the manuscript. Did the flow rate was adjusted to the membrane composition? The flow rate can influence the thickness of the membrane.

- Term swelling degree should be changed for uptake degree. Swelling refers to dilation or volumetric expansion.

- Did the PV experiments were performed at constant temperature? In the discussion section, there are no results related to the influence of temperature. It is well known that increases in temperature of PV experiments caused decreasing the separation factor with simultaneously increasing evaporation flux. 

- Separation factor should be expressed using the beta, not alpha. Alpha represents selectivity which is related to the ratio of permeability.

- Could you provide me with the membrane diameter and selective membrane area? The flux is high so I would like to know how big the membrane was.

- Lines 303-305. The sentence starts with "Moreover". This sentence is misleading. SD increased in comparison with the membrane modified by 2 g of MXene. But still was lower in comparison with pristine PVA based membrane. Moreover, can authors explain how inhomogeneous distribution and agglomeration influence uptake degree? Is this related to the rigidification of polymer chains around fillers or interfacial voids around fillers? This information would be helpful. 

- Contact angles of water. Did the authors compare the contact angle with the surface roughness of the membranes? It is well known that surface roughness influence on the value of contact angle. 

- Line 352. Permeate side should be used instead of the permeation side.

- Line 353. The driving force in pervaporation is a difference between the chemical potential feed and permeate side. The difference in concentration across the membrane results in a continuous, smooth gradient in the chemical potential of the component across the membrane.

- Line 356. It should be the feed side instead of the feeding side.

- Line 356 - 359.  This sentence is overestimated. Thirst of all word dissolution should be changed for the sorption. Interaction during the first step (sorption) of the solution-diffusion mechanism could be estimated by using Hansen's Solubility Parameters (HSP) and calculating the delta parameter based on HSP. In these calculations modified HSP parameters should be used (taking into account incorporated fillers and crosslinkers). Without these calculations, it can not be estimated that the incorporation of the fillers increases the interaction between the membrane and separated components during the sorption step. 

- Figure 9. What does this dashed line represent? It looks like Robeson's upper-bound line (which is only used in gas separation). If the authors want to compare various results, the isospines (line with the constant value of the Pervaporation Separation Index - PSI) line should be used. Isopsines are useful tools for evaluating the efficiency of various membranes. But in this case, the total flux should be used.

Author Response

Point 1: Why did the authors not prepare membranes using a constant casting solution flow rate and PTFE support winding rate? The range of flow and the winding rate is listed in the manuscript. Did the flow rate was adjusted to the membrane composition? The flow rate can influence the thickness of the membrane.

Response 1: Thanks for your suggestion. Initially, we thought that the parameters of the spraying should be kept in confidentiality due to our industrial development of PVA membrane. However, as a published article, we should make all the details of the experiment public. And the exact parameters of the flow rate and winding rate can be see in the Line 135 and 137 of our revised mauscript.

Point 2: Term swelling degree should be changed for uptake degree. Swelling refers to dilation or volumetric expansion.

Response 2: Thank you for your suggestion. Generally, swelling degree, which have been used in the articles widely, was a very commen characterization of the PVA pervaporation membranes [1-4].

[1] Rhim, J. W., Yoon, S. W., Kim, S. W., & Lee, K. H. (1997). Pervaporation separation and swelling measurement of acetic acid‐water mixtures using crosslinked PVA membranes. Journal of applied polymer science, 63(4), 521-527.

[2] Praptowidodo, V. S. (2005). Influence of swelling on water transport through PVA-based membrane. Journal of molecular structure, 739(1-3), 207-212.

[3] Yuan, H. K., Ren, J., Ma, X. H., & Xu, Z. L. (2011). Dehydration of ethyl acetate aqueous solution by pervaporation using PVA/PAN hollow fiber composite membrane. Desalination, 280(1-3), 252-258.

[4] Svang-Ariyaskul, A., Huang, R. Y. M., Douglas, P. L., Pal, R., Feng, X., Chen, P., & Liu, L. (2006). Blended chitosan and polyvinyl alcohol membranes for the pervaporation dehydration of isopropanol. Journal of Membrane Science, 280(1-2), 815-823.

Point 3: Did the PV experiments were performed at constant temperature? In the discussion section, there are no results related to the influence of temperature. It is well known that increases in temperature of PV experiments caused decreasing the separation factor with simultaneously increasing evaporation flux.

Response 3: Thank you for your question. The exact testing temperature of the PV experiment was 40 ℃ and was added in the Line 187 of the revised manuscript.

Point 4: Separation factor should be expressed using the beta, not alpha. Alpha represents selectivity which is related to the ratio of permeability.

Response 4: Thanks for you advise. In the research field of pervaporation membrane materials, alpha was always used to express the separation factor [1-5].

[1] Guo, R., Hu, C., Li, B., & Jiang, Z. (2007). Pervaporation separation of ethylene glycol/water mixtures through surface crosslinked PVA membranes: Coupling effect and separation performance analysis. Journal of Membrane Science, 289(1-2), 191-198.

[2] Rhim, J. W., & Kim, Y. K. (2000). Pervaporation separation of MTBE–methanol mixtures using cross‐linked PVA membranes. Journal of applied polymer science, 75(14), 1699-1707.

[3] Huang, R. Y. M., & Yeom, C. K. (1990). Pervaporation separation of aqueous mixtures using crosslinked poly (vinyl alcohol)(PVA). II. Permeation of ethanol-water mixtures. Journal of membrane science, 51(3), 273-292.

[4] Peters, T. A., Poeth, C. H. S., Benes, N. E., Buijs, H. C. W. M., Vercauteren, F. F., & Keurentjes, J. T. F. (2006). Ceramic-supported thin PVA pervaporation membranes combining high flux and high selectivity; contradicting the flux-selectivity paradigm. Journal of Membrane Science, 276(1-2), 42-50.

[5] Huang, R. Y. M., & Yeom, C. K. (1991). Pervaporation separation of aqueous mixtures using crosslinked polyvinyl alcohol membranes. III. Permeation of acetic acid-water mixtures. Journal of membrane science, 58(1), 33-47.

Point 5: Could you provide me with the membrane diameter and selective membrane area? The flux is high so I would like to know how big the membrane was.

Response 5: Yes, absolutely. The diameter of membrane sheet for testing was 4.5 cm and the selective membrane area was 15.89 cm2.

Point 6: Lines 303-305. The sentence starts with "Moreover". This sentence is misleading. SD increased in comparison with the membrane modified by 2 g of MXene. But still was lower in comparison with pristine PVA based membrane. Moreover, can authors explain how inhomogeneous distribution and agglomeration influence uptake degree? Is this related to the rigidification of polymer chains around fillers or interfacial voids around fillers? This information would be helpful.  

Response 6: Firstly, we should confirm that SD of the PVA-based membrane was opposited with the water resistance. That is to say, membrane with low SD will have high water resistance. Hence, just as we explained in the Lines 356 to 362, polymeric PVA matrix could be crosslinked by the GC molecules and MXene nanosheets. Both of them reinforced the PVA-based membrane matrix and made the membrane structure stable later on, which resisted the swelling of the PGM series membrane and endowed them lower SD than the PVA membrane.

When the amount of MXene added was excessive, the originally dispersed MXene nanosheets will aggregate, reducing the contact area between MXene nanosheets and PVA molecular chains. This will decrease the crosslinking effect of MXene nanosheets as crosslinking points on PVA molecular chains, ultimately resulting in an increase in SD.

Of course, the addition of MXene nanosheets can cause the PVA molecular chains to become rigid and create a porous structure at the interface, but these are not the reasons for the decrease in SD.

Point 7: Contact angles of water. Did the authors compare the contact angle with the surface roughness of the membranes? It is well known that surface roughness influence on the value of contact angle.

Response 7: Thanks for your suggestion. We agreed that water contact angle of the membrane was affected by its surface roughness. However, water contact angle (WCA) is the most direct evidence for characterizing the surface hydrophilicity of a membrane. In this study, the influence of the addition of PVA, GC, and MXene on the surface hydrophilicity of the membrane was investigated by comparing water contact angles, rather than exploring the reasons for changes in WCAs. Factors that affect the WCA of the PVA membrane surface mainly include surface roughness and surface chemical structure. Generally speaking, when the surface is a hydrophilic surface, that is, when there are a large number of hydrophilic functional groups on the membrane surface, the larger the surface roughness, the greater the contact area between the membrane surface and water droplets, resulting in a smaller water contact angle and better hydrophilicity. When the surface is a hydrophobic surface, the larger roughness of the membrane surface will increase the water contact angle of the membrane. Therefore, according to the above analysis, an increase in the content of PVA and MXene brings a large number of hydrophilic functional groups to the membrane, which reduces the water contact angle of the membrane, while the crosslinking agent GC consumes the hydroxyl functional groups in the membrane, thereby increasing the water contact angle of the membrane. In addition, MXene as an inorganic nanoparticle can also increase the surface roughness of the membrane, further reducing the water contact angle of the membrane. The decrease in water contact angle indicates an increase in the hydrophilicity of the membrane, which will be beneficial to improving the permeability performance of the PVA membrane.

Point 8: Line 352. Permeate side should be used instead of the permeation side.

Response 8: Thanks for your advice. We have corrected it to permeation side in the Figure 7 (Line 440)

Point 9: Line 353. The driving force in pervaporation is a difference between the chemical potential feed and permeate side. The difference in concentration across the membrane results in a continuous, smooth gradient in the chemical potential of the component across the membrane.

Response 9: During the process of pervaporation testing, the ethanol concentration in the raw material liquid was 95%, and the water content was only 5%. If the driving force of permeation was only the concentration gradient, water molecules will not actively move from the feed side of the membrane to the permeation side. According to classical dissolution diffusion model, the PVA membrane is a hydrophilic dense membrane, and the solubility of water molecules in the PVA membrane is higher than that of ethanol molecules in the raw material liquid. At the same time, the high vacuum on the permeation side reduces the concentration of substances on the raw material side of the membrane, creating a large concentration gradient in the PVA membrane from the feed side to the permeation side. As a result, a large number of water molecules, which are preferentially dissolved, will pass through the PVA molecular chains and diffuse to the permeation side of the PVA membrane, achieving the enrichment of water molecules on the permeation side of the membrane. Therefore, although the driving force for the permeation of water molecules in the PVA membrane is based on the concentration gradient generated by the high vacuum on the permeation side, this concentration gradient also has the same permeation effect on ethanol molecules and is not selective. The unique hydrophilic molecular structure of the PVA membrane is the main reason for the dehydration of ethanol solution.

Point 10: Line 356. It should be the feed side instead of the feeding side.

Response 10: Thanks for your advice. We have corrected it to feed side in the Line 440 (Figure 7) and Line 448

Point 11: Line 356 - 359.  This sentence is overestimated. Thirst of all word dissolution should be changed for the sorption. Interaction during the first step (sorption) of the solution-diffusion mechanism could be estimated by using Hansen's Solubility Parameters (HSP) and calculating the delta parameter based on HSP. In these calculations modified HSP parameters should be used (taking into account incorporated fillers and crosslinkers). Without these calculations, it can not be estimated that the incorporation of the fillers increases the interaction between the membrane and separated components during the sorption step.

Response 11: Thank you for your suggestion. The first step of dissolution is adsorption, and from the perspective of the dissolution-diffusion model, the dissolution process includes the adsorption process of water molecules and ethanol molecules on the membrane surface. The Hansen Solubility parameters have been calculated and listed the results in the Lines 207-219, Lines 374-377, Lines 390-391 and Line 450

Point 12: Figure 9. What does this dashed line represent? It looks like Robeson's upper-bound line (which is only used in gas separation). If the authors want to compare various results, the isospines (line with the constant value of the Pervaporation Separation Index - PSI) line should be used. Isopsines are useful tools for evaluating the efficiency of various membranes. But in this case, the total flux should be used.

Response 12: Thanks for your advice. We have deleted the dashed line in the Figure 9 (Line 491).

Reviewer 2 Report

This manuscript reported a series of crosslinked PVA hollow fiber membranes with the addition of MXene fabricated by ultrasonic spray. The ethanol dehydration performances of these membranes were further evaluated by the PV process. The contents and results are of great interest to the membrane community, while there exist some major flaws in the discussion and introduction part. Based on these, I would recommend a major revision is needed to be published in “Membranes”. Please find the comments below:

1.       If my understanding is correct, the PVA/GC/MXene layer is coated on PTFE support in this work. Did the authors directly use composite membranes for material characterization or fabricate self-standing membranes only for material characterization? Please report this information since it matters to all material characterization results.

2.       The authors stressed the benefits of ultrasonic spray in the abstract, introduction, and conclusion. However, there is zero evidence ( “figure S2 is not here”) in this manuscript to support it.

3.       In lines 56 -62, the authors claimed the chemical crosslinking consumes -OH groups and then inhibits PVA swelling, which reduces both the flux and separation factor of all PVA-based membranes.

a.       First of all, generally, crosslinked PVA tends to be more selective because i) the free volume elements are more likely to be shrunken, and ii) PVA chains’ mobility is restricted and hence suppresses the deterioration of selectivity due to swelling.

b.       Secondly, assuming this sentence is true, the authors still chose chemical crosslinking in this work.

c.       More important, in line 332, “After crosslinking with the GC, the crosslinked PVA network was constructed. The free volume of the membrane matrix and the hydrophilicity decreased. It led to the fall in the water flux but increase in the separation factor.” These results support a) perfectly

4.       There is a lot of information missing in the experimental description section. For example, the concentration of PVA/GC solution and MXene suspension, the XPS experiment details, and the exact test temperature in PV (30-70C is a very wide range). In addition, GC is very efficient and also very famous for crosslinking PVA. Therefore, with so high PVA/GC ratios and high temperature (boiling water), the so-call “PVA/GC solution” is almost certain to be gelled in contained after cooling down. So how can it be still mixed with MXene suspension and sprayed on PTFE support?

5.       The Ti3C2Tx does not seem to contain fluorine, so where the C-F group in the FTIR figure comes from?

6.       How MXene crosslink with PVA? Can authors provide more proof for it? In addition, please add the chemical structures or cross-linking scheme.

7.       Equation 7 seems to be wrong and should be (1-y)/y/((1-x)/x) = (1-y)*x/(y*(1-x)) (https://link.springer.com/referenceworkentry/10.1007/978-3-662-44324-8_302). Please double-check.

8.       There are two tables 1 in this manuscript

9.       The bulges in figure 2c seem smaller than the length or width of the nanosheets shown in Figure 1, and why the authors thought it is caused by the nanosheets? Moreover, please provide the SEM images for other membranes, since the aggregation issue at high MXene loading and thicker selective layers were assumed to be the reason for the bad performance in section 3.3.

10.   In Figure 3b, both GC crosslinking and the addition of MXene have little effect on the intermolecular distance of PVA chains? Can the authors provide the XRD results of the one with higher GC and MXene amounts?

11.   The readers cannot conclude the benefit of the presence of MXene from XPS results since there is only one membrane represented here. Also did not detect fluorine from XPS? (comment 5 again)

12.   Line 306 “As to the WCA, values of the PGM series were lower than those of the PVA and PG membranes, indicating the enhancement of the membrane hydrophilicity”. Enhanced hydrophilicity?

13.   Figure 6, why does increasing PVA lead to an enhanced elongation at break and tensile strength?

14.   In figure 9, what is the test parameter (temperature, vacuum degree, and feed composition) of other literature data?

Author Response

Point 1: If my understanding is correct, the PVA/GC/MXene layer is coated on PTFE support in this work. Did the authors directly use composite membranes for material characterization or fabricate self-standing membranes only for material characterization? Please report this information since it matters to all material characterization results.

Response 1: Thanks for your advice. The objective of this study is to prepare PVA pervaporation composite membranes that can be used in industrial applications. Therefore, all the samples characterized in this paper are PVA/GC/MXene/PTFE composite membranes.

Point 2: The authors stressed the benefits of ultrasonic spray in the abstract, introduction, and conclusion. However, there is zero evidence ( “figure S2 is not here”) in this manuscript to support it.

Response 2: Thank you for your suggestion. The manuscript and support information, two word files, have been upload in the “Membranes”. Figure S2 was a very important figure in our support information. We will check it and guarantee the success of file upload in our resubmitting.

Figure S2. Two different types of gas dynamic spray nozzles, (a) WA-101 and (b) TOF-30-1.5 (Anest-Iwata. Co. Japan), were tried in our spraying processes. (c) ultrasonic spraying nozzle (FSW-6001-L, Funsonic Co. Ltd. China) used in the preparation of the membranes. (d), (e) and (f) were the sprayed fine droplets of the PVA-based solutions on the glass plates after the solutions were puffed out from the WA-101, TOF-30-1.5 and FSW-6001-L, respectively.

Point 3: In lines 56 -62, the authors claimed the chemical crosslinking consumes -OH groups and then inhibits PVA swelling, which reduces both the flux and separation factor of all PVA-based membranes.

  1. First of all, generally, crosslinked PVA tends to be more selective because i) the free volume elements are more likely to be shrunken, and ii) PVA chains’ mobility is restricted and hence suppresses the deterioration of selectivity due to swelling.
  2. Secondly, assuming this sentence is true, the authors still chose chemical crosslinking in this work.
  3. More important, in line 332, “After crosslinking with the GC, the crosslinked PVA network was constructed. The free volume of the membrane matrix and the hydrophilicity decreased. It led to the fall in the water flux but increase in the separation factor.” These results support a) perfectly

Response 3: Thanks for your questions. Firstly, we acknowledged that the thermal and chemical cross-linking of PVA membranes are important pathways to prevent swelling and maintain structural stability. However, it is undeniable that the thermal self-crosslinking process, in particular, depletes hydroxyl functional groups of PVA molecular chains, resulting in a decrease in the overall hydrophilicity of the PVA membrane, making it more compact and less favorable for the dissolution and diffusion of water molecules. The chemical cross-linking agent, glycerol, used in this study, has three hydroxyl functional groups. Apart from consuming two hydroxyl functional groups to cross-link with PVA molecular chains, one un-crosslinked hydroxyl functional group is left, which maintains the number of hydrophilic functional groups in the membrane to a certain extent. In addition, compared with thermal self-crosslinking, glycerol as a cross-linking agent increases the spacing between PVA molecular chains, avoiding excessive compaction of the PVA membrane structure. However, as a small molecule, glycerol still cannot significantly improve the performance of the PVA membrane. MXene nanosheets have the highest affinity for water molecules, which can significantly increase the number of hydrophilic functional groups in cross-linked PVA membranes. At the same time, the organic-inorganic interfacial interface between MXene nanosheets and PVA molecules constructs rapid permeation micro-channels for water molecules in the PVA membrane matrix. Therefore, the prepared PGM mixed matrix membrane has the best permeation and vaporization performance.

Point 4: There is a lot of information missing in the experimental description section. For example, the concentration of PVA/GC solution and MXene suspension, the XPS experiment details, and the exact test temperature in PV (30-70C is a very wide range). In addition, GC is very efficient and also very famous for crosslinking PVA. Therefore, with so high PVA/GC ratios and high temperature (boiling water), the so-call “PVA/GC solution” is almost certain to be gelled in contained after cooling down. So how can it be still mixed with MXene suspension and sprayed on PTFE support?

Response 4: Thank you for your question. The composition of different PVA/GC/MXene composite membranes were all listed in Table 1. Then, the concentration of PVA/GC solution and MXene suspension can be calculated by it.

But I’m sorry that we forgot to supplement the XPS experiment details, which have been added in our revised manuscript (Lines 150 to 151).

The exact testing temperature of PV was 40℃,which has been listed in the Line 187.

Firstly, our PVA polymer should be absolutly dissolved in the boiling water. After the PVA/GC solution being cooled down, the solution still have excellent fluidity and can be mixed with MXene suspension, because the highest solid loading of PVA were only 10wt% (as shown in Table 1). Moreover, the PVA-based solution were always heated to 40℃ before spraying when it was in the feeding tank and delivery pipe (Line 134), which was aimed to smooth transfer of the solution and spraying.

Point 5: The Ti3C2Tx does not seem to contain fluorine, so where the C-F group in the FTIR figure comes from?

Response 5: MXene nanosheets were fabricated by the in-situ etching method which was provided in our previous study. [1]

[1] Liu, Q.; Pan, X.; Xu, N.; Wang, Q.; Qu, S.; Wang, W.; Fan, L.; Dong, Q. Hypergravity field induced self-assembly of 2D MXene in polyvinyl alcohol (PVA) membrane matrix and its improvement of alcohol/water pervaporation. J. Appl. Polym. Sci. 2023, e53740.

Point 6: How MXene crosslink with PVA? Can authors provide more proof for it? In addition, please add the chemical structures or cross-linking scheme.

Response 6: Thanks for your suggestion. We have analyzed the FTIR in detail that MXene nanosheets in the membrane mainly achieve cross-linking with PVA molecular chains through hydrogen bonding interactions between their surface Ti-OH and C-OH groups in PVA molecular chains, as well as Ti-O-C covalent bonds. GC molecules mainly achieve cross-linking with PVA molecular chains through covalent bonds. The supplemental figures and investigation words can be seen in the Lines 265 to 281.

Point 7: Equation 7 seems to be wrong and should be (1-y)/y/((1-x)/x) = (1-y)*x/(y*(1-x)) (https://link.springer.com/referenceworkentry/10.1007/978-3-662-44324-8_302). Please double-check.

Response 7: Thank you very much. We have checked it and revised the equation six.

Point 8: There are two tables 1 in this manuscript

Response 8: Thank you very much. We have revised it.

Point 9: The bulges in figure 2c seem smaller than the length or width of the nanosheets shown in Figure 1, and why the authors thought it is caused by the nanosheets? Moreover, please provide the SEM images for other membranes, since the aggregation issue at high MXene loading and thicker selective layers were assumed to be the reason for the bad performance in section 3.3.

Response 9: Thanks for you suggestion. EDX mapping images of C, O and Ti elements in the PVA and PGM-0 membranes’ top-surface was supplemented in the supplemental information (Figure S3). And the revised part was added into the Lines 243 to 245 of the revised manuscript. The solid content of PVA is positively correlated with the thickness of the membrane, and it is a well-known fact that increasing the membrane thickness leads to an increase in permeation resistance, which has been repeatedly demonstrated in membrane separation research. As an inorganic nanoparticle, MXene has a tendency to aggregate at the lowest surface energy. Therefore, there should be an optimal amount of MXene added, beyond which MXene will inevitably aggregate like other nanomaterials, leading to defects in the membrane structure and a decrease in membrane performance.

Point 10: In Figure 3b, both GC crosslinking and the addition of MXene have little effect on the intermolecular distance of PVA chains? Can the authors provide the XRD results of the one with higher GC and MXene amounts?

Response 10: Thank you for your suggestion. The cross-linking of GC and MXene with PVA membrane would definitely affect the crystal structure of the PVA membrane. As shown in Figure 3c, the characteristic peaks of PVA become more prominent with the addition of GC and MXene. As you mentioned, the cross-linking of GC and MXene brings the PVA molecular chains closer together. It can be inferred that as the amount of GC and MXene added increases, the characteristic peaks of the PVA membrane will become more prominent. Therefore, we believe that it does not seem necessary to add additional XRD data for further explanation.

Point 11: The readers cannot conclude the benefit of the presence of MXene from XPS results since there is only one membrane represented here. Also did not detect fluorine from XPS? (comment 5 again)

Response 11: The purpose of providing XPS data in the manuscript is to demonstrate that MXene and GC have been successfully added to the PVA membrane and that they have formed a new chemical structure with PVA molecules, resulting in cross-linking of PVA molecules. The advantage of adding MXene can be demonstrated by water contact angle, thermogravimetric curve, and PV data. MXene is prepared by in-situ HF etching method, and the presence of fluorine atoms on its surface has been confirmed by numerous studies. The C-F structure was detected in the infrared data, but not in the XPS data. The possible reason is that in the peak separation of C element, the peak area of C-C, C-OH, C-Ti, and C=O are all of high intensity, which leads to the C-F structure not being clearly visible in the peak separation.

Point 12: Line 306 “As to the WCA, values of the PGM series were lower than those of the PVA and PG membranes, indicating the enhancement of the membrane hydrophilicity”. Enhanced hydrophilicity?

Response 12: Water contact angle is the most direct data to characterize the hydrophilicity of a membrane. A decrease in contact angle indicates an increase in the membrane's hydrophilicity.

Point 13: Figure 6, why does increasing PVA lead to an enhanced elongation at break and tensile strength?

Response 13: With the increase of PVA content, the polymer chain density in the PVA matrix will also increase. This increase in chain density can lead to the formation of more intermolecular hydrogen bonds, making the mechanical properties of the PVA matrix stronger.

Point 14: In figure 9, what is the test parameter (temperature, vacuum degree, and feed composition) of other literature data?

Response 14: We chose these references for comparison because they all reported on high-performance ethanol/water pervaporation separation membranes within the past 5 years. When designing our experiments, we considered the impact of testing parameters on performance. Therefore, our testing temperature, vacuum degree, and feed composition were kept consistent with other major reference papers.

Reviewer 3 Report

In this research, the authors tried to prepare PVA mixed matrix composite membrane for the PV process of ethanol dehydration. The article highly shows the research of a composite membrane based on PVA. Comparisons with known membranes based on PVA are given. However, due to insufficient experimental results and common separation performance of the obtained membranes, I cannot recommend this article to be publishable in this form. Authors should cite experiments depending on the ethanol/water content. Base on the main comments mentioned above, in my opinion, the manuscript needs major revision before publication. There are some other comments as follow:

1. How about the stability of the membrane under long-term testing?

2. How does the membrane perform under different flow conditions?

3. It should be given in detail about the substrate on which the polymer composition was applied.

Author Response

Point 1: How about the stability of the membrane under long-term testing?

Response 1: Thanks for your question. The longest continuous stable operating time of the PVA-based pervaporation membrane that we have prepared so far is 182 days.

Point 2: How does the membrane perform under different flow conditions?.

Response 2: Thank you for your question. The pervaporation process is different from liquid separation membrane processes such as microfiltration, ultrafiltration, nanofiltration, and reverse osmosis. The flow state and pressure of the feed solution have little effect on the pervaporation performance, while the latter is more sensitive to concentration polarization and membrane fouling, and is therefore more affected by the flow state of the feed solution. As a result, research on pervaporation has rarely addressed the study of the flow state of the feed solution.

Point 3: It should be given in detail about the substrate on which the polymer composition was applied.

Response 3: Thank you for your question. In the manuscript, we have provided detailed information on the relevant parameters of the supporting material, which is made of PTFE with an average pore size of 0.22 μm, an average thickness of 200 μm, a width of 350 mm, and a length of 30 m per roll. (as shown in the Lines 137 to 138)

Round 2

Reviewer 3 Report

Dear Author,

The additional result and discussion help to understand the common readers. Paper can now be accepted.